# Insomnia among Prison Officers and Its Relationship with Occupational Burnout: The Role of Coping with Stress in Polish and Indonesian Samples

**DOI:** 10.3390/ijerph18084282

**Published:** 2021-04-17

**Authors:** Ewa Sygit-Kowalkowska, Andrzej Piotrowski, Imaduddin Hamzah

**Affiliations:** 1Department of Psychology, Kazimierz Wielki University, 85-867 Bydgoszcz, Poland; esygit@ukw.edu.pl; 2Institute of Psychology, University of Gdańsk, 80-309 Gdańsk, Poland; 3Community Guidance, Politeknik Ilmu Pemasyarakatan, Depok 16514, Indonesia; imaduddin@poltekip.ac.id

**Keywords:** insomnia, job burnout, prison officers, coping with stress

## Abstract

Thus far, data on sleep disorders among prison officers (POs) have been scarce. Research allows us to relate this problem to occupational stress, which POs experience every day. The aim of the current study was to analyze the scale, predictors, and impact of select factors on the relationship between insomnia and occupational burnout. This study was carried out on a sample of 376 Indonesian and 288 Polish POs using the Athens Insomnia Scale (AIS), the Coping Orientation to Problems Experienced (COPE) inventory, and the Oldenburg Burnout Inventory (OLBI). Results showed that 43.4% of the Polish sample exhibited early symptoms of insomnia, compared to 26.1% of the Indonesian sample. Sleep disorders had a significant role in developing occupational burnout. In both samples, coping strategies such as help-seeking and engagement were revealed to have a mediating role in the relationship between insomnia and occupational burnout dimensions. For the total sample and for the Polish sample, the coping strategy of help-seeking was the only predictor of insomnia. Discrepancies (concerning the role of age, gender, and multi-shift work) were observed between the current results and earlier studies. The current study’s limitations were discussed and new solutions were proposed.

## 1. Introduction

Sleep is necessary for human functioning. The latest research shows that around one-third of the general population suffers from some form of sleep disorder. However, among penitentiary personnel, this figure is over 40% [1,2]. Sleep disorders negatively impact functioning and health [3,4]. They generate anxiety, which significantly reduces work effectiveness [5,6,7]. The most common sleep disorder is insomnia. Insomnia is defined as unsatisfactory, interrupted sleep; difficulties in falling asleep and maintaining sleep; poor quality of sleep; or sleep that is not restorative and which negatively influences functioning throughout the day [8]. Insomnia is related to numerous negative health consequences [9], for example, fatigue and irritability, impaired daytime functioning, and mood disorders. The etiology of insomnia is multifaceted and includes the subjective stress experience. An analysis of data from insomnia patients showed that work-, family-, and health-related stressors are the most frequently related to insomnia [10]. Among work-related stressors, conflicts with supervisors, interpersonal difficulties, overwork, and financial difficulties are emphasized as influencing the development of insomnia, together with such stressors as difficulties in workplace relationships, insufficient support in the workplace, and low job satisfaction [11,12]. Sleep disorders, including insomnia, also occur frequently in post-traumatic stress disorder (PTSD) [13], which can be a consequence of traumatic experiences related to various occupations, for example, soldiers experiencing active combat, prison officers (POs) experiencing assault by inmates, or rescue personnel responding to natural and manmade disasters [14,15].

Poor sleep quality is also related to numerous physical and mental health problems, which has been documented in workplace and organization psychology research. Sufficiently meeting the need for sleep facilitates professional activity. The literature notes that one of the factors of the development and maintenance of insomnia is multi-shift work [16,17,18,19,20]. In the 1997 International Classification of Sleep Disorders, dyssomnia (disorders related to the abnormal length and quality of sleep) included shift work-related sleep disorders or stress-induced sleep disorders. Data show that shift work and the number of work hours negatively impact the experience of workplace stress, sleep, wakefulness, and fatigue [11,21]. Night shift workers also report lower sleep quality and experience more problems with falling asleep and maintaining sleep compared to daytime workers [22,23]. Research on nurses shows a high incidence of low sleep quality, a negative impact of the number of night shifts per week, and a beneficial effect of sleeping 2–3 h during night shifts [24,25].

The negative impact of low sleep quality on the health and workplace effectiveness of various occupational groups has been shown to be so severe that discontinuing research in this direction is unjustifiable. Moreover, the results provoke discussions on possible solutions aiming to improve sleep quality among night shift workers, particularly those who are responsible for public safety, as low sleep quality is related to a higher frequency of workplace errors [26].

### 1.1. Occupational Burnout and Insomnia

Workplace conditions have a documented effect on employee health. Depending on their characteristics, this effect can be positive or negative [27]. If workplace conditions are negative, they may lead to occupational burnout syndrome. This happens when long-term exposure to emotionally difficult workplace situations facilitates physical, emotional, and mental exhaustion [28]. Occupational burnout is the result of chronic workplace stress which the individual has not adequately coped with. The co-occurrence of stressors and ineffective coping behaviors negatively impacts motivation and engagement in goal realization, which translates into lower workplace effectiveness.

The most popular theory of occupational burnout, developed by Maslach and colleagues, defines it as a syndrome comprised of emotional exhaustion, depersonalization, and a reduced sense of personal accomplishment [29]. It has been shown that occupational burnout syndrome is a frequent problem and it has been tied to significant costs, related to employee demoralization, alienation, and exhaustion [30]. Occupational burnout models have been modified throughout the years. New diagnostic questionnaires have been developed and symptom descriptions have been adjusted. For example, Demerouti’s concept of measuring burnout involves the two dimensions of employee energy and identification, which allow results to be interpreted in terms of both burnout and work engagement.

A consistent relationship between occupational burnout and insomnia has been established [31,32]. The literature indicates that it has a two-way character. This means that both occupational burnout and insomnia allow the prediction of each’s development and intensity over time [33]. One can be a risk factor of the other [31]. This is proven by studies showing that employees exhibiting high levels of occupational burnout also report lower sleep quality and greater mental exhaustion [33,34]. The literature assumes that sleep disorders caused by stress might lead to physical and mental exhaustion, which is one of the symptoms of occupational burnout [33,35]. The risk of occupational burnout is greater among employees suffering from insomnia, lack of restful sleep, and higher anxiety levels. Sleep quality, alongside such factors as organizational climate and job satisfaction, is indicated as one of the determinants of occupational burnout [36]. Thus, this confirms the strong relationship between insomnia and occupational burnout [37]. However, some data show that insomnia increases the risk of occupational burnout symptom persistence (including emotional burnout), although occupational burnout in itself does not influence the development of insomnia [38,39].

Analogously to studies on insomnia, analyses of occupational burnout prevalence have been carried out on medical personnel and school staff [40,41,42,43]. In the studies on penitentiary officers carried out thus far, occupational burnout has been analyzed in relation to organizational stressors [44]. However, the effects of stress manifested through such symptoms as insomnia have not been studied in this context.

The specifics and nature of a given job might constitute a significant burden for the employees. These are often the constant features of the job, for example, time pressure in medical professions or strict security measures in the uniformed services. Research points towards new possibilities of application, including significant improvements for employees. Thus far, it has been shown that personality factors influencing the development of occupational burnout include high anxiety, low resilience, external locus of control, avoidant style of coping with stress, low self-esteem, and high job motivation, among others [45].

### 1.2. Prison Service: Characteristics of Work in the Polish and Indonesian Penitentiary Systems

Penitentiaries are closed and guarded facilities for individuals serving prison sentences. They are also the job environment of penitentiary personnel who are responsible for executing temporary arrests and enforcing appropriate restrictions, carrying out interventions directed at the inmates, and guaranteeing their laws and appropriate living conditions. The characteristic aspects of prison service workers are: working in a uniformed formation, an authoritarian style of management, a paramilitary nature, formalism of activity, and risk of aggression on the part of the inmates. These specifics are sources of stress. Thus far, it has been shown that stressors also include both social contacts between the POs as well as interactions with the inmates [46,47]. Empirical analyses confirm that high levels of overwork (in the context of perceived job demands) and low independence predict psychological tension and physical illnesses among penitentiary personnel [48]. Prison officers report that such aspects of their work as punishments for absences, pressure from the management, understaffing, fear of letting their colleagues down, employment instability, and a sense of duty and professionalism facilitate presenteeism, or going to work while ill [49]. Eighty-four percent of one sample reported such behaviors (including 53% reporting their frequency as “always”). Thus, it is not surprising that studies of penitentiary personnel show that the longer the job experience, the stronger the turnover intentions [50].

Penitentiary personnel experience the mental and physical effects of stress. This influences their judgments of life and job satisfaction. Occupational stress and burnout are also more frequent among penitentiary personnel than the general population [51,52,53,54,55]. Additionally, hypertension is more frequently diagnosed among penitentiary personnel, and they relate it to the characteristics of their workplace [56].

Measurements of sleep disorders among all penitentiary personnel (POs, administration workers, social workers, technicians, healthcare workers, and managers) showed a higher incidence compared to other occupational groups [57]. Therapeutic POs and their work ability, measured by the Work Ability Index (WAI), were significantly related to anxiety levels, and partially to loss of sleep [58]. Regarding prison officers, insomnia has been studied in relation to health problems using the General Health Questionnaire (GHQ-28) rather than specific insomnia measures [59]. Thus far, the majority of analyses of insomnia indexed in the EBSCO database (located using the keywords: “insomnia or sleep disorders or sleep disturbance” AND “prison officer or prison guard or correctional officer or prison staff”) have concerned insomnia among inmates and pharmacological treatments. Penitentiary personnel are almost entirely ignored, both in the context of sleep disorders and otherwise.

Studies also show differences in health dependent on the occupation within penitentiary facilities. Compared to healthcare, administration, or socioeducational workers in penitentiaries, prison guards responsible for maintaining safety experience more mood and anxiety disorders. Years of job experience were also related to the intensity of mental health symptoms in a sample of French penitentiary personnel [57]. Personnel working directly with inmates experienced the highest number of stressors. On the other hand, stressors were the lowest in the group of administration employees, where contact with inmates is minimal or nonexistent [60]. Studies have also shown that shiftwork among penitentiary personnel impacts their family life. Night shifts can cause greater fatigue and stress in this occupational group [61]. Literature reviews also indicate results showing that employee experiences vary depending on the security level of their facility (semi-open or closed). Prison officers in closed prisons report significantly higher occupational stress [62], stating that the most important stressors included exhaustion, traumatic events, and favoritism. In high-security prisons, job experience and age were related to the experience of stressors.

As was already shown, penitentiary personnel are at risk of experiencing occupational burnout. Studies carried out in China indicate that such factors as job stress, job dangerousness, or male gender influence the intensity of occupational burnout symptoms. On the other hand, role clarity reduces burnout symptoms [63]. Moreover, occupational burnout among POs is one of the strongest predictors of depressive symptoms [64]. Identified consequences of occupational burnout include addiction behaviors such as alcohol consumption, smoking, and substance abuse. Risk of violence and occupational burnout are significant for the development of addiction behaviors in this occupational group [65,66].

There is a well-founded view in psychology that studying stress should also include studying individual behaviors in stressful situations. Thus far, studies on penitentiary personnel have shown that employees who frequent religious services experience mood disorders less frequently [67]. Using social support was also negatively correlated with occupational burnout and stress [68,69]. Simultaneously, longer job experience as a PO is significantly related to passive coping behaviors which inhibit problem-oriented coping [70].

The functioning of penitentiary systems is related to the directions of national criminal policy. In turn, social pathologies are related to socioeconomic conditions [71]. The current study analyzed two democratic countries: Poland, an EU member state (ranking 50th on the democracy index in 2020) and Indonesia, an Asian country (ranking 64th) [72]. Many differences in these countries’ characteristics warrant such an analysis. On the Human Development Index, comprising such variables as life expectancy, years of education, and gross national income per capita, Poland ranks 35th and Indonesia ranks 107th [73]. However, there are also further arguments warranting a comparative analysis of these countries. November 2020 data from the Institute for Crime and Justice Policy Research show that the prison population (per 100,000 people) was 91 in Indonesia and 180 in Poland [74]. In Poland, the prisoner per prison staff proportion was 2.46 in 2020, whereas in Indonesia, it was over three times larger. Both countries’ prison services employ around four times more men than women. In Indonesia, serious assaults occurred 4.18 per 100,000 people in 2018, whereas in Poland, they were over four times more frequent. The robbery rate in Poland was 17.9 in 2018, and in Indonesia, it was over six times lower. The Indonesian penitentiary system struggles with such problems as terrorism, overcrowding (occupancy level of 183.9%), the death penalty, or drug-related crimes [75,76,77]. In Poland, overcrowding in prisons is consistently being reduced (occupancy level of 84%), the death penalty has been abolished, and terrorism has not occurred. However, Polish POs report that their job is rarely satisfactory [78] and it is not respected by the general public [79]. This stands in contrast to Indonesia, where prison officers earn higher wages than police officers or soldiers.

Few studies have concerned insomnia among both countries’ populations, including those occupational groups who are tasked with protecting citizens’ safety and realizing other basic functions of the state. Study results suggest that self-reported insomnia is a common problem in the general Polish population. The prevalence of sleep complaints was 50.5% (58.9% in women, 41.4% in men) among group of 2413 participants [80]. A study of 2924 respondents based on data from the Indonesian Family Life Survey (2014–2015) found that over 47% stated that they had moderate sleep disorders [81]. Another study among Indonesian workers found a significant positive relationship between sleep disorders and subjectively reported fatigue [82]. Reference searches on the main website for scientific publications in Indonesia (http://garuda.ristekbrin.go.id/ (accessed on 4 February 2021) with the keyword “insomnia” show that studies were mainly conducted among students and the elderly. Studies of insomnia or sleep disturbances among workers are very limited. No studies among POs have been found, either in Poland or Indonesia. Moreover, Indonesia lacks specialist facilities dedicated to PO mental health. General health facilities, including hospitals, can be accessed through health insurance. However, in Poland, specialist Mental Health Centers for the Prison Service offer individual therapy, crisis interventions, therapy for PTSD, and personnel assessment and recruitment.

The aim of the current study was to examine the phenomenon of insomnia and its relationship with occupational burnout in a sample of Polish and Indonesian POs. The analysis included:-The role of strategies of coping.-The role of organizational factors, that is, penitentiary unit type (open, closed, semi-open) and single- or multi-shift work.-The role of individual factors such as gender and age.

The following research questions were formulated:-What is the intensity of occupational burnout among POs? Are there differences between the Polish and Indonesian samples?-Do coping strategies, organizational factors, and individual factors differentiate insomnia levels?-Are there differences between the samples?-Which of the considered variables are predictors of insomnia intensity among POs in each sample?

## 2. Materials and Methods

### 2.1. Ethics Approval

The study was carried out in compliance with the Declaration of Helsinki and the standards of the Ethical Code of the Polish Psychological Association. The study was approved by the Ethics committee of the Faculty of Psychology, Kazimierz Wielki University in Bydgoszcz, Poland, on 28 February 2020. Consent was also given by the managers of each penitentiary facility taking part in the study, as well as by their supervisory bodies: The District Inspectorate of the Prison Service in Poland and the Directorate General of Corrections, Ministry of Law and Human Rights, Republic of Indonesia.

The POs’ participation in the study was voluntary and anonymous. Questionnaires were placed in commonly accessible rooms. The questionnaires were supplemented with a description of the study and its aims, as well as a written request to fill them out and insert them in a specially prepared container. The description also informed the participants that the questionnaires were entirely anonymous and their results would only be used for the purpose of the current study. The data collection method resulted in a sample which was not random, since the questionnaires were only filled out by interested participants. However, this data collection method was optimal because it allowed the participants’ full anonymity and privacy to be maintained.

The following measures were used:(1)The Athens Insomnia Scale (AIS) by Soldatos, Dikeos, and Paparrigopoulos. This is an eight-item scale measuring insomnia symptoms [83]. The scale is a self-report method comprising eight items related to various symptoms of insomnia. It is most frequently used for diagnostic purposes as well as to measure therapeutic effectiveness. The AIS has high reliability and validity. Each item is rated on a four-point Likert-type scale, where 0 denotes a lack of a given symptom and 3—its significant intensity. The assessed symptoms are: sleep induction, awakenings during the night, waking up in the morning, total sleep duration, sleep quality, wellbeing during the day, mental and physical functioning capacity the next day, and sleepiness during the day. The total score is between 0 and 24, where higher scores denote lower sleep quality. The original criteria consider scores above six as reliable indicators of early symptoms of insomnia [83,84]. The higher the scores, the more severe the symptoms [85,86]. The internal reliability of the AIS was 0.88 in the Polish sample and 0.74 in the Indonesian sample.(2)The Oldenburg Burnout Inventory (OLBI) by Demerouti and Bakker [87]. This questionnaire represents a two-factor model of occupational burnout. It consists of 16 items forming two subscales:
(a)Exhaustion due to intense physical, mental, and emotional effort, the effect of chronic workplace stress related to job demands;(b)Distancing from work (described as disengagement from work by Demerouti and Bakker), its purpose, and its character [88].

Answers are given on a four-point Likert-type scale, where 1 denotes “I agree” and 4—“I disagree”. The sum of the item scores divided by their number gives the total subscale scores in a range of 1–4. The higher the scores, the higher the severity of the two components of occupational burnout—exhaustion and distancing from work. The Polish part of the study used the adaptation of the OLBI by Baka and Basińska (2016) [89]. The Cronbach’s α reliability coefficient in the adaptation study was 0.73 for the exhaustion subscale and 0.69 for the disengagement subscale. The Indonesian part of the study used the adaptation by Santoso and Hartono (2017) [90]. The Cronbach’s α reliability coefficients were 0.798 and 0.753 for the exhaustion and disengagement subscales, respectively.

The Cronbach’s α reliability coefficient for the Polish and Indonesian versions of the exhaustion subscale was 0.73 and 0.80, respectively, and 0.69 and 0.75, respectively, for the distancing subscale.
(3)Coping Orientation to Problems Experienced (COPE) by Carver, Scheier, and Weintraub [91].

This questionnaire measures strategies of coping with stress. It is a self-report measure consisting of 60 items. Responses are given on a four-point Likert-type scale, where 1 denotes “I usually do not do this at all” and 4—“I usually do this a lot”. The questionnaire distinguishes 15 coping strategies. The current study used a three-factor classification [90], according to which the coping strategies can be divided into:The engagement coping style: acceptance, positive reinterpretation, active coping, planning, restraint coping, and suppression of competing activities.The disengagement coping style: religion, mental disengagement, behavioral disengagement, and denial.The help-seeking style: seeking instrumental support, seeking emotional support, and venting of emotions.

The Polish part of the study used the adaptation of the COPE by Juczyński and Ogińska-Bulik (2009) [92]. The Cronbach’s α reliability coefficients for the individual subscales ranged from 0.48 to 0.94. The reliability of the subscales in the Polish sample, measured with Cronbach’s α, was 0.82 for the engagement coping style, 0.65 for the disengagement coping style, and 0.50 for the help-seeking coping style. In the Indonesian sample, these were 0.86, 0.46, and 0.71, respectively. For the purpose of the Indonesian part of the study, the COPE inventory was translated from English to Bahasa Indonesia, and then back-translated to English. We compared the resulting version of the questionnaire with the English-language original in order to maintain consistency. Thus far, in Indonesia, this questionnaire has been used in adolescent and young adult samples. Reliability testing of this questionnaire revealed that the Cronbach’s α reliability coefficients ranged between 0.5 and 0.9. [93,94,95]. The COPE inventory was used in accordance with the information included on the COPE creator’s website: “You are welcome to use all scales of the COPE, or to choose selected scales for use (see below regarding scoring). Feel free as well to adapt the language for whatever time scale you are interested in” [96]. The study also included a demographic questionnaire collecting the following data: penitentiary unit type (open, closed, semi-closed), education, gender, and age. The participants were also asked whether they work in the multi-shift system.

### 2.2. Data Analysis

Data analysis was carried out using IBM SPSS version 25.0 (IBM SPSS, Armonk, NY, USA). The software was used to analyze basic descriptive statistics and the normality of distribution. To compare the groups on their quantitative variable scores, independent-samples Student’s *t* test or Mann–Whitney’s *U* test were used. To estimate the relationships between the quantitative variables, Pearson’s correlation analysis was carried out. To compare the correlation coefficients in the two groups, Fisher’s test was used. To examine the mediating role of coping strategies in the relationship between insomnia and occupational burnout, a mediation analysis was carried out using the PROCESS macro 3.5 [97]. The final stage of the analysis involved a path analysis using the AMOS software (version 24.0) (IBM SPSS, Chicago, IL, US). to identify the predictors of insomnia. α = 0.05 was taken as the statistical significance level for the analyses.

## 3. Results

### 3.1. Descriptive Statistics

A total of 778 people took part in the study: 288 from Poland (PL) and 376 from Indonesia (I). Sixty (I, PL) participants worked in semi-open facilities, and 316 (I) and 217 (PL) worked in closed facilities. Three Polish participants worked in open facilities. The descriptive statistics of subjects are shown in Table 1.

The first step of the analysis involved calculating the basic descriptive measures together with the Kolmogorov–Smirnov normality of distribution test. The analysis showed that neither of the analyzed variables approached normality. Nevertheless, considering the sample size and the skewness values being within the −2;2 range, it can be assumed that the deviation from the normal distribution was not significant. The descriptive statistics are shown in Table 2.

### 3.2. Insomnia among Prison Officers: Differences between the Countries

Mean insomnia levels among the POs were *M* = 5.45 (*SD* = 4.32). The distribution of this variable was right-skewed, which means that the majority of the results were above the sample average.

Next, using the independent-samples *t* test, insomnia levels among Polish and Indonesian POs were compared. The analysis showed statistically significant differences between the groups, *t*(465.22) = 10.54; *p* < 0.001; *d* = 0.94; *95%CI* [2.76; 4.02]. Insomnia levels were higher among the Polish (*M* = 7.50; *SD* = 5.11) than the Indonesian POs (*M* = 4.11; *SD* = 3.07). An analysis using the cut-off point for early symptoms of insomnia on the AIS showed that 26.1% of the Indonesian sample and 43.4% of the Polish sample met the cut-off point.

### 3.3. The Relationship between Coping Styles and Insomnia Levels

To assess the relationship between coping styles and insomnia levels, a Pearson’s correlation analysis was carried out. The analysis was carried out for the total sample, as well as separately for the Polish and Indonesian samples. The analysis showed a weak, negative relationship between insomnia levels and engagement in the total sample, as well as a weak, negative relationship between insomnia levels and help-seeking. This means that the more frequent the use of engagement and help-seeking coping styles, the lower the insomnia levels.

An analysis using Fisher’s test to compare the correlation coefficients showed statistically significant differences between Polish and Indonesian POs in terms of engagement and help-seeking. The intergroup differences in disengagement were not statistically significant. In the Polish sample, insomnia levels were weakly and negatively correlated with engagement and help-seeking. No relationship between insomnia levels and disengagement was observed. In the Indonesian sample, the only statistically significant relationship occurred between insomnia levels and disengagement. It was weak and positive, which means that the higher the frequency of using this coping style, the higher the insomnia levels. Detailed results of the analyses are shown in Table 3.

### 3.4. Organizational Factors and Insomnia Levels

Using Mann–Whitney’s *U* test, insomnia levels were compared between the POs working in a multi-shift system and those who did not. The analysis showed that multi-shift workers in the total sample reported higher insomnia levels. Analogous differences were observed in the Polish sample. The effect size for these differences was weak. In the Indonesian sample, the differences in insomnia levels between POs working in the multi-shift system and those who did not were not statistically significant. The results are presented in Table 4.

Additionally, Polish and Indonesian POs’ insomnia levels were compared separately for those working in the multi-shift system and those working single shifts. The analysis showed that both among multi-shift workers and among single-shift workers, higher insomnia levels were reported by Polish POs. The effect size for the differences was moderate. The results of the analysis are shown in Table 5.

Next, analogous analyses were carried out for POs working in semi-open and closed facilities. The analysis did not show statistically significant differences in insomnia levels between the facility types, both for the total sample (*p* = 0.266, r = 0.04) as well as within the Polish (*p* = 0.857, r = 0.01) and Indonesian samples (*p* = 0.624, r = 0.02). Nonetheless, insomnia levels among Polish POs, both those working in semi-open and closed facilities (*M* = 7.00; *M* = 7.00, respectively) were higher than among Indonesian POs in those facility types (*M* = 3.00; *M* = 3.00).

### 3.5. Sociodemographic Factors and Insomnia Levels

The next step of the analysis focused on the relationship between insomnia levels and sociodemographic factors. The analyses included the participants’ gender and age.

To estimate the differences in insomnia levels between women and men, Mann–Whitney’s *U* test was used. The analysis showed that both in the total sample as well as among the Polish prison officers, men exhibited higher insomnia levels than women. In the Indonesian sample, the gender differences were not statistically significant. Both Polish female and male POs had higher insomnia levels than Indonesian female and male POs (see Table 6).

The relationship between age and insomnia was not statistically significant in the total sample (*r* = 0.03; *p* = 0.341). Fisher’s test showed statistically significant differences between the samples in the relationship between age and insomnia (*Z* = 3.01; *p* = 0.003). A weak and positive relationship between age and insomnia was observed in the Polish sample (*r* = 0.17; *p* = 0.003). The higher the Polish POs’ age, the higher their insomnia levels. On the other hand, in the Indonesian sample, this relationship was not statistically significant (*r* = −0.05; *p* = 0.274).

### 3.6. The Mediating Role of Coping Styles in the Relationship between Insomnia and Occupational Burnout

To examine whether coping styles (engagement, disengagement, and help-seeking) are mediators in the relationship between insomnia levels and occupational burnout, a mediation analysis was carried out using Hayes’ PROCESS macro 3.5 (2017). To estimate the statistical significance of the indirect effects, the analyses were carried out using the bootstrapping method with a sampling of 5000 to estimate the percentile confidence intervals for the effects. When the confidence interval between the upper and lower bounds was above 0, the effect was not statistically significant. However, when it did not cross the 0 value, that is, when both bounds of the confidence interval were positive and negative, the effect was statistically significant. A total of six models were analyzed. The detailed results of the analyses are discussed below.

**Model** *1:*
*The mediating role of engagement in the relationship between insomnia and exhaustion.*


The first analyzed model considered the mediating role of engagement for the relationship between insomnia and exhaustion as a dimension of occupational burnout. The model had a good fit to data, *F*(2.783) = 177.54; *p* < 0.001 and it explained 31.2% of the variance in exhaustion. Figure 1 shows the analyzed model.

The analysis showed a negative relationship between insomnia and engagement (*b* = −0.13; *SE* = 0.02; β = −0.20; *p* < 0.001; Path A). As insomnia increased by one unit of measurement, engagement decreased by 0.13 units. The direct relationship between insomnia and exhaustion was positive (*b* = 0.05; *SE* < 0.01; β = 0.49; *p* < 0.001; Path C)—as insomnia increased by one unit, exhaustion increased by 0.05 units. The relationship between engagement and exhaustion with the inclusion of insomnia (*b* = −0.05; *SE* < 0.01; β = −0.28; *p* < 0.001; Path B) was negative—the higher the engagement, the lower the exhaustion. The relationship between insomnia and exhaustion with the inclusion of engagement (*b* = 0.05; *SE* < 0.01; β = 0.43; *p* < 0.001; Path C’) was also statistically significant and positive—as insomnia increased, together with including the coping strategies, exhaustion increased by 0.05 units.

To test whether the indirect effect of engagement was statistically significant, an additional bootstrapping analysis was carried out. It showed that the effect was significant (*b* = 0.01; BootSE < 0.01; BootLLCI = 0.004; BootULCI = 0.009). This means that engagement weakened the positive relationship between insomnia and exhaustion, though it remained statistically significant (partial mediation).

**Model** *2:*
*The mediating role of engagement in the relationship between insomnia and distancing from work.*


The second analyzed model included the mediating role of engagement in the relationship between insomnia and distancing from work as a dimension of occupational burnout. The analyzed model had a good fit to data, *F*(2.783) = 154.10; *p* < 0.001, and it explained 28.2% of the variance in distancing. Figure 2 shows the model.

The analysis showed a negative relationship between insomnia and engagement (*b* = −0.13; *SE* = 0.02; β = −0.20; *p* < 0.001; Path A). As insomnia levels increased by one unit of measurement, engagement levels lowered by 0.14 units. The direct relationship between insomnia and distancing was positive (*b* = 0.05; *SE* < 0.01; β = 0.46; *p* < 0.001; Path C)—as insomnia levels increased by one unit of measurement, distancing increased by 0.05 units. The relationship between insomnia and distancing with the inclusion of engagement was also statistically significant and positive (*b* = 0.05; *SE* < 0.01; β = 0.40; *p* < 0.001; Path C’)—as insomnia levels increased, with the inclusion of coping styles, distancing increased by 0.05 units.

To test the statistical significance of the indirect effect, an additional bootstrapping analysis was carried out. It showed that the effect was significant (*b* = 0.01; BootSE < 0.01; BootLLCI = 0.004; BootULCI = 0.010). This means that engagement weakened the positive relationship between insomnia and distancing, although it remained statistically significant (partial mediation).

**Model** *3:*
*The mediating role of disengagement in the relationship between insomnia and exhaustion.*


The third analyzed model included the mediating role of disengagement for the relationship between insomnia and exhaustion as a dimension of occupational burnout. The analyzed model had a good fit to data *F*(2.783) = 127.28; *p* < 0.001, and it explained 24.5% of the variance in exhaustion. Figure 3 shows the model.

The analysis showed a lack of relationship between insomnia and disengagement (*b* = −0.02; *SE* = 0.01; β = −0.04; *p* = 0.245; Path A).

The direct relationship between insomnia and exhaustion was positive (*b* = 0.05; *SE* < 0.01; β = 0.49; *p* < 0.001; Path C)—as insomnia levels increased by one unit of measurement, exhaustion increased by 0.05 units. The relationship between disengagement and exhaustion with the inclusion of insomnia was negative (*b* = −0.03; *SE* = 0.01; β = −0.09; *p* = 0.006; Path B)—the higher the disengagement, the lower the exhaustion. The relationship between insomnia and exhaustion with the inclusion of disengagement as a coping style was statistically significant and positive (*b* = 0.05; *SE* < 0.01; β = 0.48; *p* < 0.001; Path C’)—as insomnia increased, with the inclusion of disengagement, exhaustion increased by 0.05 units of measurement.

To test the statistical significance of the indirect effect of coping styles, an additional bootstrapping analysis was carried out. It showed that the effect was not statistically significant (*b* < 0.01; BootSE < 0.01; BootLLCI = −0.001; BootULCI = 0.001). This means that disengagement was not a significant mediator in the relationship between insomnia and exhaustion.

**Model** *4:*
*The mediating role of disengagement in the relationship between insomnia and distancing from work.*


The fourth model included the mediating role of disengagement in the relationship between insomnia and distancing from work as a dimension of occupational burnout. The analyzed model had a good fit to data, *F*(2.783) = 138.93; *p* < 0.001, and it explained 26.2% of the variation in distancing. Figure 4 shows the model.

The analysis showed a lack of relationship between insomnia and disengagement (*b* = −0.02; *SE* = 0.01; β = −0.04; *p* = 0.245; Path A). The direct relationship between insomnia and distancing from work was positive (*b* = 0.05; *SE* < 0.01; β = 0.46; *p* < 0.001; Path C)—as insomnia levels increased by one unit of measurement, distancing increased by 0.05 units. The relationship between disengagement and distancing with the inclusion of insomnia was negative (*b* = −0.07; *SE* = 0.01; β = −0.23; *p* < 0.001; Path B)—the higher the disengagement, the lower the distancing. The relationship between insomnia and distancing with the inclusion of disengagement was positive (*b* = 0.05; *SE* < 0.01; β = 0.45; *p* < 0.001; Path C’)—as insomnia increased, with the inclusion of disengagement, distancing lowered by 0.05 units.

To test the statistical significance of the indirect effect of disengagement, an additional bootstrapping analysis was carried out. It showed a lack of statistical significance of the indirect effect (*b* < 0.01; BootSE < 0.01; BootLLCI = −0.001; BootULCI = 0.003). Disengagement was not a significant mediator in the relationship between insomnia and distancing from work.

**Model** *5:*
*The mediating role of help-seeking in the relationship between insomnia and exhaustion.*


The fifth analyzed model included the mediating role of help-seeking in the relationship between insomnia and exhaustion as a dimension of occupational burnout. The analyzed model had a good fit to data, *F*(2.783) = 168.82; *p* < 0.001, and it explained 30.1% of the variance in exhaustion. Figure 5 shows the model.

The analysis showed a negative relationship between insomnia and help-seeking (*b* = −0.10; *SE* = 0.01; β = −0.25; *p* < 0.001; Path A). As insomnia levels increased by one unit of measurement, the use of the help-seeking coping strategy decreased by 0.10 units. The direct relationship between insomnia and exhaustion was positive (*b* = 0.05; *SE* < 0.01; β = 0.49; *p* < 0.001; Path C)—as insomnia levels increased by one unit of measurement, exhaustion increased by 0.05 units. The relationship between help-seeking and exhaustion with the inclusion of insomnia was negative (*b* = −0.07; *SE* = 0.01; β = −0.26; *p* < 0.001; Path B)—the higher the use of help-seeking as a coping strategy, the lower the exhaustion. The relationship between insomnia and exhaustion with the inclusion of help-seeking as a coping strategy was positive (*b* = 0.05; *SE* < 0.01; β = 0.42; *p* < 0.001; Path C’)—as insomnia levels increased, with the inclusion of help-seeking, exhaustion increased by 0.05 units.

To test the statistical significance of the indirect effect of help-seeking, an additional bootstrapping analysis was carried out. It showed that this effect was statistically significant (*b* = 0.01; BootSE < 0.01; BootLLCI = 0.005; BootULCI = 0.010). This means that help-seeking was a significant mediator in the relationship between insomnia and exhaustion. Additionally, the relationship between the explanatory and the explained variables remained statistically significant—a partial mediation effect.

**Model** *6:*
*The mediating role of help-seeking in the relationship between insomnia and distancing from work.*


The last analyzed model included the mediating role of active coping in the relationship between insomnia and distancing from work as a dimension of occupational burnout. The analyzed model had a good fit to data, *F*(2,783) = 164.68; *p* < 0.001, and it explained 29.6% of the variance in distancing. Figure 6 shows the model.

The analysis showed a negative relationship between insomnia and help-seeking (*b* = −0.10; *SE* = 0.01; β = −0.25; *p* < 0.001; Path A). As insomnia levels increased by one unit of measurement, the frequency of using this coping strategy decreased by 0.10 units. The direct relationship between insomnia and distancing was positive (*b* = 0.05; *SE* < 0.01; β = 0.46; *p* < 0.001; Path C)—as insomnia levels increased by one unit of measurement, distancing increased by 0.05 units. The relationship between help-seeking and distancing with the inclusion of insomnia was negative (*b* = −0.09; *SE* = 0.01; β = −0.31; *p* < 0.001; Path B)—the higher the frequency of using the help-seeking coping style, the lower the distancing. The relationship between insomnia and distancing with the inclusion of help-seeking was statistically significant and positive (*b* = 0.04; *SE* < 0.01; β = 0.38; *p* < 0.001; Path C’)—as insomnia increased, with the inclusion of the help-seeking coping strategy, distancing increased by 0.05 units of measurement.

To test the statistical significance of the indirect effect of help-seeking, an additional bootstrapping analysis was carried out. It showed that the effect was statistically significant (*b* = 0.01; BootSE < 0.01; BootLLCI = 0.006; BootULCI = 0.012). This means that the coping strategy of help-seeking weakened the relationship between insomnia and distancing, although it remained statistically significant (partial mediation).

### 3.7. Predictors of Insomnia

To examine the relationships between the coping strategies, organizational and sociodemographic factors, and insomnia levels, a path analysis was carried out using the maximum likelihood (ML) estimation. The analysis included a total of 751 cases after excluding missing data.

After including additional covariations between the variables on the basis of the modification indexes, the analyzed model achieved a good fit to data, χ^2^/df = 2.65; CFI = 0.976; RMSEA = 0.047; SRMR = 0.047, and it explained a total of 7% of the variance in insomnia levels. A detailed analysis showed a lack of statistically significant relationships between insomnia and age (β = 0.04; *p* = 0.290), penitentiary unit type (β = −0.03; *p* = 0.407), engagement (β = −0.03; *p* = 0.628), and disengagement (β = 0.06; *p* = 0.158). Significant relationships were observed between insomnia and gender, help-seeking, and multi-shift work. Help-seeking (β = −0.22; *p* < 0.001) and multi-shift work (β = −0.10; *p* = 0.007) were negatively related with insomnia, meaning that the higher the frequency of using help-seeking as a coping strategy, the lower the insomnia levels, while multi-shift work increased insomnia levels. Gender (β = 0.08; *p* = 0.027) was positively related to insomnia, which indicates that men experienced higher insomnia levels. Figure 7 shows the standardized regressions coefficients for the model.

To test whether the analyzed model differed depending on the sample country, additional analyses were carried out separately for the Polish (*n* = 288) and Indonesian (*n* = 376) samples. To compare the models between the groups, the software created by Gaskin and Lim (2018) was used [98]. It uses the differences between the chi-squared values and tests the intergroup differences. The analysis showed statistically significant differences between the models for the Polish and the Indonesian samples, Δχ^2^ = 19.24; *df* = 7; *p* = 0.007; ΔNFI = 0.018. Thus, it can be assumed that the intergroup differences for the analyzed model are significant. Table 7 shows a comparison of the standardized values for both samples’ models.

The analysis showed that both groups differ with respect to the relationship between age and insomnia as well as between help-seeking and insomnia. In the Polish sample, the relationship between age and insomnia was positive and statistically significant, while in the Indonesian sample, it was not statistically significant. In the Polish sample, the relationship between help-seeking and insomnia was negative and statistically significant, while in the Indonesian sample, it was not statistically significant. The differences between the models for the remaining relationships were not significant. The model for the Polish sample explained 3% of the variance in insomnia levels, while the model for the Polish sample explained 4%. Figure 8 shows both models.

## 4. Discussion

Analyses of sleep disorders in difficult and dangerous professions indicate that they frequently occur among the uniformed services, and that police officers and firefighters often report waking up too early and sleeping at work [99]. Insomnia, one of the most common forms of sleep disorders, can worsen physical and psychological functioning, leading to the development of depression, increasing the risk of suicide attempts, alcohol, and substance abuse [100]. Sleep disorders impair executive functions, leading to more frequent mistakes at work [101]. Thus, it is warranted to continue researching sleep disorders. Thus far, data on the prevalence of insomnia among POs have been scarce, and studies have largely focused on inmates.

The current study showed that a high proportion of penitentiary personnel exhibit early symptoms of insomnia. This is an important finding, especially considering the job demands faced by POs, because insomnia is related to lower mental health and social functioning [102]. Insomnia symptoms in the Polish sample were almost four times higher than in a sample of Polish professional drivers and nurses, also measured with the AIS [103,104]. Although the mean results for the total sample in the current study were lower than those reported by oncological patients, the difference was 3% for the current Polish sample, with POs reporting higher insomnia scores [105].

In the current study, men reported higher insomnia levels. Male gender was also one of the predictors of insomnia in the total sample. The results of the Indonesian sample were in line with previous Indonesian studies in showing that gender does not have a statistically significant influence, although younger men (15–34 years) report higher insomnia levels than older men (45+ years) [106,107]. However, to explain the current results, the nature of work in the penitentiary system should be taken into account. This leads to fewer female employees. Relatively few female penitentiary personnel have direct contact with inmates. Comparative data are lacking, although studies on inmates in Great Britain using the Sleep Condition Indicator showed that 70.6% of sleep disorder cases were women [108]. Notably, meta-analyses on gender differences also indicate that sleep disorders predominantly affect women [109]. Similar differences also involve age. On the one hand, AIS results among firefighters and rescue workers are related to age [110]. This confirms the current results among POs. On the other hand, there are also data showing that the incidence of insomnia, diagnosed using the DSM-IV-TR criteria, does not increase with age (*N* = 3970). However, these data come from a phone interview study, which limits their interpretation [111].

The current study also confirmed the relationship between sleep disorders and occupational burnout, including physical, mental, and emotional exhaustion, and distancing from work. These results are in line with previous studies [112]. Thus far, the issue of sleep disorders among prison officers was not taken up in the literature alongside the relationship between occupational stress and burnout. The causal relationship between occupational stress and sleep disorders was already studied in the 1990s, and it continues to find confirmation in more recent studies [113,114]. Current studies indicate that a low level of control at work (due to the nature of penitentiary work) might facilitate the experience of stress and the development of insomnia [115]. The literature reviewed earlier and—indirectly—the current results lead to the conclusion that stress might be a common element between occupational burnout and sleep disorders. Many studies show that POs’ physical and mental health is lower than among other occupational groups, which is explained by reference to the experience of chronic stress at work [78,116]. Moreover, general physical health impacts sleep quality [117]. A stressful, unpredictable work environment, insomnia (lack of restful sleep), sensitivity to stress, and difficulties with coping (occupational burnout) create a vicious circle. Its subsequent negative effects are caused by prior ones. Moreover, it has been shown that individuals exhibiting high insomnia scores engage in health behaviors less frequently [118]. Thus, the current results seem consistent.

The literature review above showed that insomnia causes numerous negative health consequences. These can impact the capacity to cope with stress and inhibit adaptive behaviors. A lack of appropriate coping skills can lead to worsened mental, social, and professional functioning in penitentiary personnel [119,120]. Dividing the COPE coping styles into three groups, the current study showed that active coping (engagement) and using social support and information seeking (help-seeking) are more frequently reported by those participants who indicated having better sleep. Active coping is considered as potentially effective and more likely to be employed by individuals with high resilience [121]. By changing the situation, active coping significantly improves quality of life [122]. Thus far, studies on insomnia have mainly analyzed stress rather than coping. However, studies on Polish probation officers indicate a significant impact of coping styles on individual dimensions of occupational burnout as measured by the Maslach Burnout Inventory (MBI), explaining between 12 and 23% of their variance [123]. A study on a sample of German students confirms the current results. It shows that lower sleep quality is related to chronic stress and less frequent problem-oriented coping and help-seeking [124]. The role of specific coping styles among POs is also in line with the results from a sample of arterial hypertension patients. Patients suffering from insomnia reported using such coping strategies as positive reframing (grouped into the Engagement coping style in the current study) and emotional support (grouped into the Help-seeking style) less frequently, and behavioral disengagement (grouped into Disengagement) more frequently [125]. Notably, the importance of the latter coping strategy has not been observed in the penitentiary context. Avoidance behaviors such as denial do not lead to problem-solving. However, the nature of penitentiary work requires a proactive approach, and avoidance of difficult situations is undesirable. Nevertheless, results of studies on adults suffering from insomnia show that denial (grouped into Disengagement) can be the only coping strategy that is a significant predictor of insomnia symptoms [126]. Analyses have shown that sleep quality is related to coping abilities. However, it is known that individual activity undertaken in response to stress is determined by personality traits such as neuroticism, ruminative tendencies, self-criticism, or depressed mood [127,128]. This can cause potential difficulties for penitentiary personnel. Data show a high incidence of depression and anxiety in this occupational group compared to other professions and to the general population [129].

The current results allow for underscoring the role of multi-shift work in the development of sleep disorders. It is known that some employees cannot adjust to multi-shift work and are thus at risk for shift work disorder (SWD) [130]. Its symptoms include sleep disorders, insomnia, or hypersomnia. The significance of this variable for the Indonesian sample has not been confirmed in the current study, even though data from this country indicate otherwise. Studies on Indonesian manual workers and nurses show that aside from age and tobacco consumption, multi-shift work impacts the development of insomnia [131,132,133]. Multi-shift work can lead to over six- or ten-times greater risk of insomnia in these two occupational groups. Moreover, sleep disorders—as has been shown in a sample of packaging terminal employees—increase the risk of workplace accidents during night shifts [134].

## 5. Conclusions

On the basis of the current study, analysis, and discussion of the results, it can be concluded that:-
Scientific advances allow occupational burnout to be linked with sleep disorders. Experiences of stress are considered to factor into the etiology of these two phenomena. Nevertheless, data on the role of coping strategies in the relationship between occupational burnout and insomnia have been scarce thus far.-Insomnia among Polish and Indonesian prison officers has not been examined thus far. The current study shows that its incidence is higher in Poland, while over 1/4 of the Indonesian sample reported results indicating early insomnia symptoms. Sleep disorders were revealed to influence the development of occupational burnout.-The mediating role of such coping strategies as help-seeking and engagement in the relationship between insomnia and occupational burnout dimensions has been confirmed for the total sample. Both in the total sample, as well as in the Polish sample, help-seeking was the only coping strategy predicting insomnia.-Age, gender, and multi-shift work differences exist between the current comparison of the two countries and the results of previous studies.-The current study should be followed up with further studies, which should include more precise measurement tools for sleep difficulties. This is because sleep disorders can significantly impact the work functioning of penitentiary personnel.

## 6. Strengths, Limitations, and Future Research

The current results can constitute a significant argument for health promotion campaigns highlighting sleep hygiene directed at penitentiary personnel. They also have potential value for the managers of penitentiary institutions who are responsible for general work safety and hygiene. Well-rested employees are more engaged. The current results also highlight the issue of multi-shift work. Organizing shift changes such that they will interfere in employee health to the least extent possible seems to be the only solution. This has not been taken into consideration thus far in the Polish and Indonesian penitentiary systems. Coping skills trainings have been shown to positively impact adaptability, facilitate more effective coping, and reduce anxiety levels among students and other nonclinical groups [135,136,137]. There is precedent for implementing solutions in this area as, for example, soft skills training is part of the training curriculum for Polish penitentiary personnel.

The limitations of the current study stem in part from its choice of diagnostic tools. Following the grouping of coping styles into three groups by Gutiérrez et al. [138], the COPE subscales of humor and substance use have been excluded from the analyses. Including them in future studies seems warranted. However, it would be necessary to consider that the majority of the penitentiary personnel in Indonesia are Muslim, and the Muslim faith prohibits alcohol use. Thus, their results on the substance use subscale could reflect their adherence to religion rather than preferred coping styles. The COPE questionnaire also allows other solutions: analyzing individual subscales or dividing the coping styles into avoidance coping, active coping, and help-seeking and emotion-focused coping. Additionally, the current study used the basic version of the diagnostic measure of insomnia, which includes a single cut-off point for early insomnia symptoms. Other measures of insomnia, such as the Insomnia Severity Scale, the sleep diary, the Pittsburgh Sleep Quality Index, or the Stanford Sleepiness Scale, allow for more precise descriptions of sleep problems. Considering the extent of sleep problems reported in the current study, it seems warranted to measure the chronotypes of penitentiary personnel. This allows to estimate the peak activity point. It has been shown that chronotype influences individual sleep patterns [139]. Thus far, such analyses have not been carried out on POs. Moreover, aside from using strictly psychological questionnaires, interdisciplinary studies using objective medical measures are also possible [140].

A research model created for the purposes of future studies would allow for measuring the frequency of health behaviors, including the general category of preventive behaviors. A study on a sample of telecommunication company employees and nurses showed that insomnia symptoms were related to alcohol use, with the highest insomnia levels being reported by high-risk drinkers, smokers, and individuals consuming more than three cups of coffee per day [141,142]. Structured interviews would also be a valuable addition. The length of the period between shifts should also be taken into consideration, since its effect has been confirmed in a sample of nurses [143]. It also seems warranted to carry out extensive studies on those penitentiary workers who report low insomnia levels, and thus have healthy sleep patterns. Employee sleepiness, together with feelings of fatigue, might drastically impair memory, learning, and attention, leading to more frequent mistakes at work [144,145].

A review of studies on treatments for insomnia shows evidence for the effectiveness of cognitive-behavioral therapy (cognitive-behavioral therapy for insomnia, CBT-I) [146]. During the pandemic period and its related organizational difficulties, it seems worthwhile to consider the creation of a smartphone app using validated solutions for self-help as well as pre- and post-intervention measurements.

## Figures and Tables

**Figure 1 ijerph-18-04282-f001:**
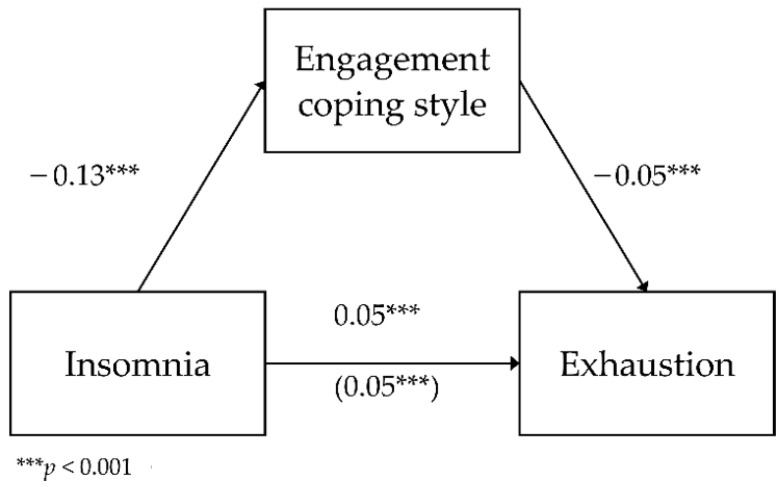
The mediating role of engagement in the relationship between insomnia and exhaustion—unstandardized coefficients.

**Figure 2 ijerph-18-04282-f002:**
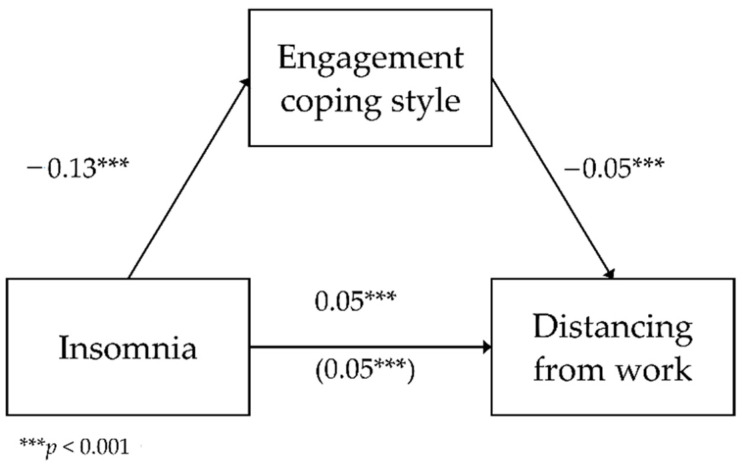
The mediating role of engagement in the relationship between insomnia and distancing from work—unstandardized coefficients.

**Figure 3 ijerph-18-04282-f003:**
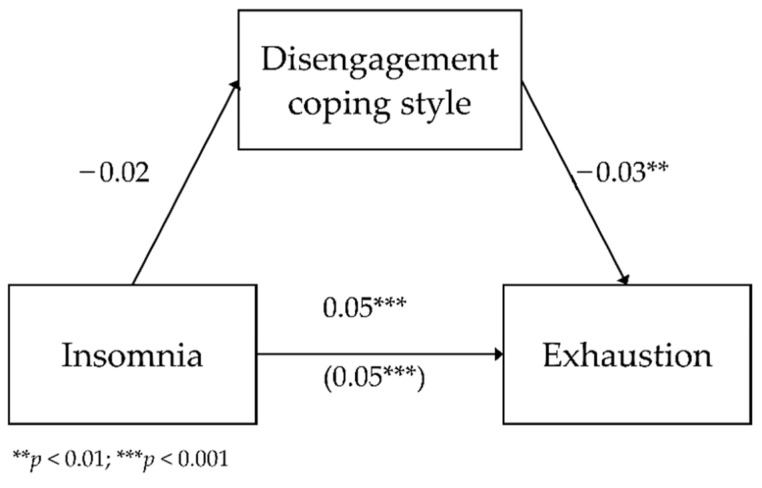
The mediating role of disengagement in the relationship between insomnia and exhaustion—unstandardized coefficients.

**Figure 4 ijerph-18-04282-f004:**
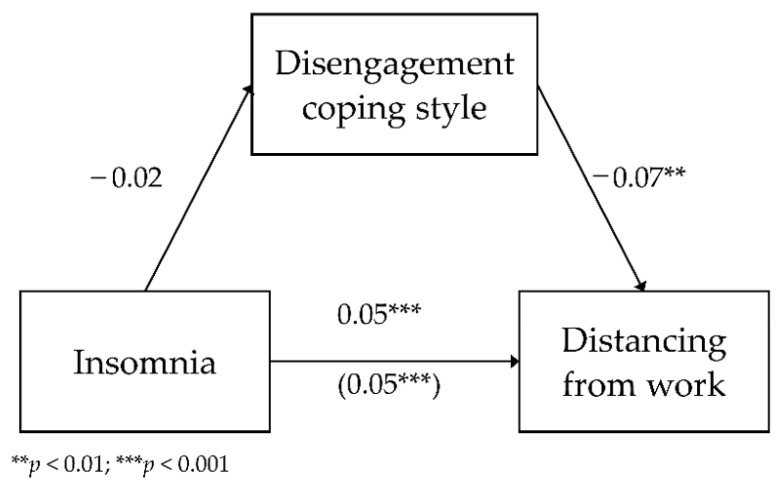
The mediating role of disengagement in the relationship between insomnia and distancing from work—unstandardized coefficients.

**Figure 5 ijerph-18-04282-f005:**
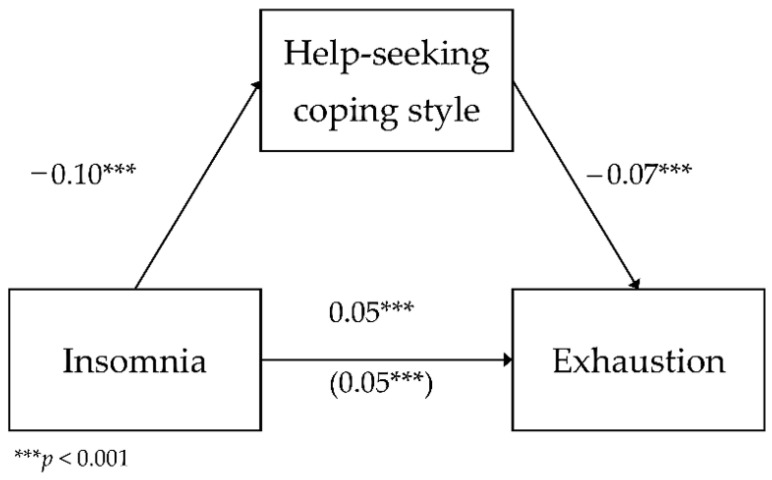
The mediating role of help-seeking in the relationship between insomnia and exhaustion—unstandardized coefficients.

**Figure 6 ijerph-18-04282-f006:**
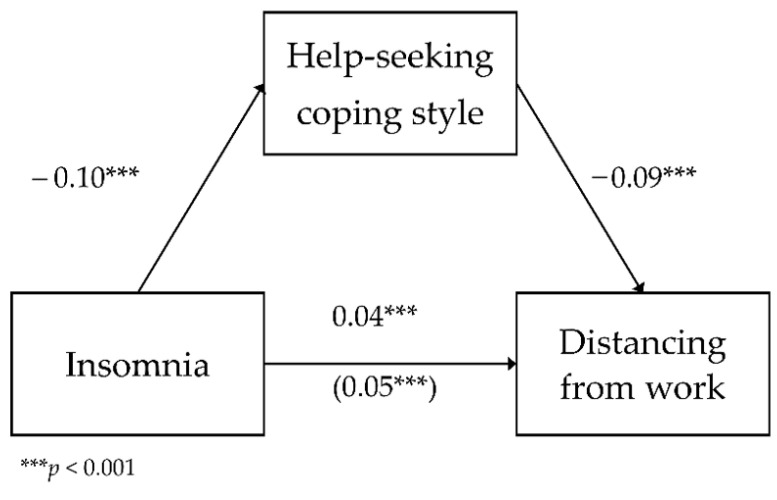
The mediating role of help-seeking in the relationship between insomnia and distancing from work—unstandardized coefficients.

**Figure 7 ijerph-18-04282-f007:**
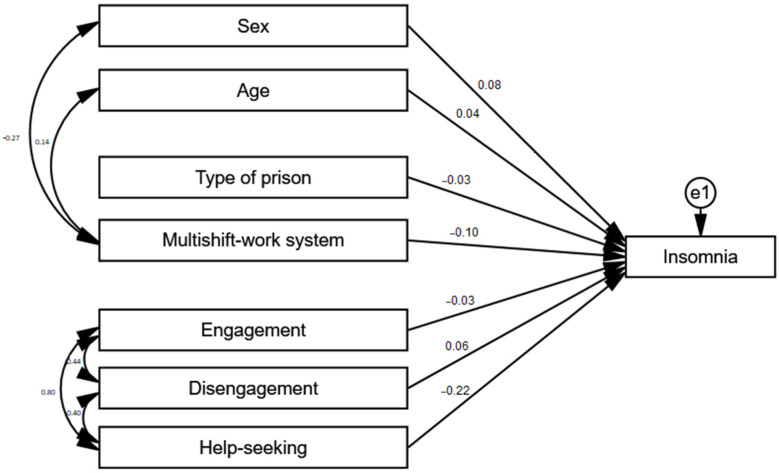
Path analysis of the explanatory model of insomnia.

**Figure 8 ijerph-18-04282-f008:**
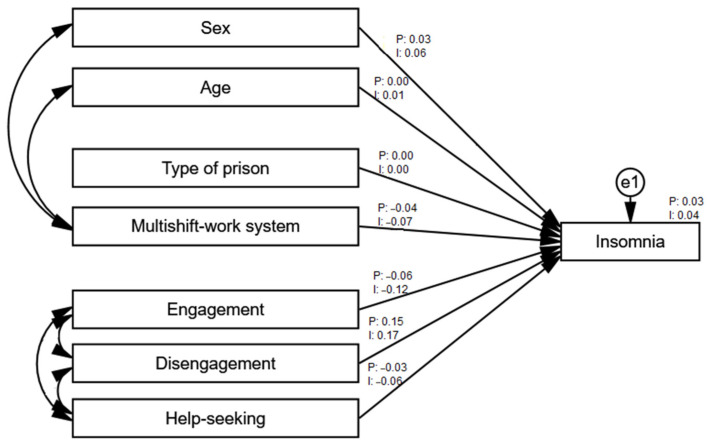
Path analysis of the explanatory model of insomnia in the Polish and Indonesian samples—standardized coefficients. Note: P—Poland, I—Indonesia.

**Table 1 ijerph-18-04282-t001:** Descriptive statistics of surveyed respondents.

	N	Age	Woman	Multi-Shift System	Secondary Education	Bachelor’s Degree	Master’s Degree	Open	Semi-Open	Closed
**Poland**	288	34.12 (SD 7.4)	33 (11.8%)	194 (68.6%)	143 (50.4%)	58 (20.4%)	83 (29.2%)	3 (1.1%)	60 (21.4%)	217 (77.5%)
**Indonesia**	376	33.61 (SD 9.4)	76 (20.2%)	175 (46.7%)	157 (41.8%)	190 (50.5%)	29 (7.7%)	0	60 (16.0%)	316 (84.0%)

SD: standard deviation; N: number of subjects.

**Table 2 ijerph-18-04282-t002:** Descriptive statistics and the normality of distribution test.

	M	Me	SD	Sc.	Kurt.	Min.	Max.	D	*p*
**Insomnia**	5.45	4.00	4.34	1.15	1.14	0.00	21.00	0.15	<0.001
**Occupational burnout**									
Exhaustion	2.28	2.25	0.48	−0.06	0.57	1.00	4.00	0.09	<0.001
Distancing	2.29	2.25	0.50	0.12	0.23	1.00	4.00	0.07	<0.001
**Coping styles**									
Engagement	16.56	16.75	2.87	−0.30	0.56	6.00	24.00	0.04	0.001
Disengagement	9.25	9.50	1.62	−0.24	0.58	4.00	15.25	0.08	<0.001
Help-seeking	8.27	8.25	1.69	−0.16	0.02	3.00	12.00	0.05	0.001

M: mean; Me: median; Sc.: Skewness, Kurt.: kurtosis; D: Kolmogorov–Smirnov test statistics; *p*: significance level.

**Table 3 ijerph-18-04282-t003:** Pearson’s correlations between insomnia levels and coping styles.

	Insomnia						Z	*P*
	Total Sample		Poland		Indonesia			
	r	*p*	r	*p*	R	*p*		
Engagement	−0.20	<0.001	−0.18	0.001	−0.03	0.575	−2.19	0.029
Disengagement	−0.04	0.245	0.09	0.126	0.12	0.010	−0.45	0.651
Help-seeking	−0.25	<0.001	−0.23	<0.001	−0.02	0.736	−2.97	0.003

r: Pearson’s correlation coefficient; Z: Fisher’s Z test statistics.

**Table 4 ijerph-18-04282-t004:** Comparison of insomnia levels between prison officers working in the multi-shift system and those who are not.

	Insomnia								
	Yes			No					
	$\overset{-}{r}$	Me	IQR	$\overset{-}{r}$	Me	IQR	Z	*P*	r
Total sample	415.30	5.00	6.00	352.57	4.00	5.00	−3.89	<0.001	0.14
Poland	156.82	7.00	8.00	123.54	7.00	7.00	−2.11	0.035	0.12
Indonesia	238.12	4.00	4.00	219.44	3.00	3.00	−1.25	0.212	0.06

$\overset{-}{r}$: average rank; IQR: interquartile range.

**Table 5 ijerph-18-04282-t005:** Comparison of insomnia levels among Polish and Indonesian prison officers working in the multi-shift system and the single-shift system.

	Poland			Indonesia					
	$\overset{-}{r}$	Me	IQR	$\overset{-}{r}$	Me	IQR	Z	*p*	r
Multi-shift system	252.71	7.00	8.00	230.87	4.00	4.00	−6.24	<0.001	0.30
Single-shift system	178.07	7.00	7.00	151.37	3.00	3.00	−6.54	<0.001	0.35

**Table 6 ijerph-18-04282-t006:** Comparison of insomnia levels between women and men.

	Women			Men					
	$\overset{-}{r}$	Me	IQR	$\overset{-}{r}$	Me	IQR	Z	*p*	R
Total sample	329.13	3.50	4.00	398.99	5.00	6.00	−3.35	0.001	0.12
Poland	123.54	5.00	7.00	156.82	7.00	7.00	−2.11	0.035	0.12
Indonesia	219.44	3.00	4.00	238.12	3.00	4.00	−1.25	0.212	0.06

**Table 7 ijerph-18-04282-t007:** Comparison of standardized regression coefficients for the paths in the Polish and Indonesian samples’ models.

X	Y		Poland			Indonesia		*Z–Score*
		B	Β	*p*	b	β	*P*	
Gender	Insomnia	1.33	0.08	0.160	0.28	0.04	0.416	−1.04
Age	Insomnia	0.10	0.14	0.011	−0.01	−0.04	0.437	−2.65 ***
Penitentiary unit type	Insomnia	<−0.01	<0.01	0.996	0.05	0.01	0.900	0.07
Engagement	Insomnia	−0.11	−0.05	0.520	−0.16	−0.15	0.068	−0.21
Disengagement	Insomnia	0.30	0.11	0.055	0.45	0.19	<0.001	0.70
Help-seeking	Insomnia	−0.64	−0.18	0.029	0.03	0.02	0.848	2.05 **
Multi-shift work	Insomnia	0.40	0.04	0.545	−0.52	−0.09	0.076	−1.28

** *p* <0.01; *** *p* < 0.001; β: standardized coefficient b.

## Data Availability

The data presented in this study are available on request from the corresponding author. The data are not publicly available due to the security practices of penitentiary institutions.

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
