# Peer review of "Insomnia among Prison Officers and Its Relationship with Occupational Burnout: The Role of Coping with Stress in Polish and Indonesian Samples"

_ijerph, 2021, doi:10.3390/ijerph18084282_

Round 1

Reviewer 1 Report

Dear Authors,

Thank you for this interesting manuscript. Below are my comments and recommendations.

Presenting results from a cross-sectional study such term as "influence" in the title should be avoided and replaced by "association" or "relationship".

The introduction is too lengthy. Please shorten it with the focus on research problems and existing gaps in the literature.

Please provide a detailed description of how study participants were recruited. Please explain the reason for recruiting prison officers from different countries. Also, the sample size calculation here is necessary.

A table with Polish and Indonesian samples' characteristics would be very helpful here.

References supporting previous validation and structure confirmation of measures used are not provided. For Burnout Inventory and Coping Orientation subscales, Cronbachs alphas are rather low and some of them are not acceptable. Some data supporting the validity of instruments used for both Polish and Indonesian translations have to be provided.

Please provide a version of the PROCESS macro used and the number of the model(s) applied.

You provide very detailed descriptive statistics for the insomnia scale, but the histogram plot is not necessary and should be removed.

Explanations of abbreviations should accompany Table 1. What is D? The tame comment is applied for tables 2-5.

Please provide a reference for insomnia cut-offs.

In Figure 8 some path coefficients are very low (e.g., for age and sex). Please indicate statistical significance in the figure. Also, I have some doubts about categorical variables (like gender) analysis with the AMOS, I think there is a problem here. You should use other software if you want to include categorical predictors like MPlus or lavaan in R (free). Next, some variables like type of prison and multishift-work system appear in the final path model. These variables are not presented in the methods section. The same comment applies if these two variables are categorical.

Overall, the paper is too lengthy. The authors should concentrate the information in the introduction, results description and discussion and reduce paper length.

Author Response

Thank you for reviewing our article, Insomnia among prison officers and its relationship with occupational burnout. The role of coping with stress in a Polish and Indonesian sample. Below, we provide information for reviewers about the specific changes we have made to the text, as well as the responses to the reviewers’ comments.

Sincerely,

The Authors

Response to Reviewer 1

  • Why use Democracy index 2107 and not 2020? Is it possible that Poland has declined in democratic values last year. At least the majority of EU-member states think so.

Thank you for this valid remark. We have corrected the information given in the text, together with Reference 93, by referring to more up-to-date data from the same source, namely:

The Economist Intelligence Unit (2021). Democracy Index 2020. In sickness and in health? https://pages.eiu.com/rs/753-RIQ-438/images/democracy-index-2020.pdf?mkt_tok=NzUzLVJJUS00MzgAAAF76GGaDaVYg-6wdWCs4pH92eTapmkx3zIQowzgyMiLbegflEd4DVCFbpwc62iBR6llXVZvrKcO8b-tVTPApA9jmu9zg8eFIYwKqp3i6FY9FwoxZA

  • There is lack of valid refence to the prevalence of insomnia. Figures in Poland authors say is 40-50%. Isn't this too high. A more clinically significant insomnia prevalence can be found in 10-15% of the population and Europe.

The prevalence of insomnia in the entire sample ranges from 10 to 15%.

The group indicated in your comment are the Polish Prison Service officers. Most respondents (almost 70%) worked in the multi-shift system, which also means working during the night. The multi-shift system involves 12-hour shifts followed by 24 hours of time off (or 48 hours in case of a night shift). The days and hours of the shifts are organized by the director of a given penitentiary unit. Because such a high proportion of prison officers work during night shifts, the prevalence of initial insomnia symptoms is high.

One of the earlier studies on a Polish sample of prison officers (Piotrowski A. Self-evaulation of personal physical health, accidents while performing duty and preventive treatment of stress in Prison Service. Medycyna Pracy. 2018;69(4):425-438. doi:10.13075/mp.5893.00535) also analyzed sleep problems (Do you have sleep disorders?). The obtained results were: sometimes – 29.6%, often – 8%, very often – 4%.

In our study, we used the AIS questionnaire to measure insomnia intensity. We used the simplest data analysis method based on the cutoff point which allows for diagnosing early symptoms of insomnia. This is also the way we described our results – we established the occurrence of significant symptoms at an early stage. They should be monitored by employers and occupational physicians.

  • Authors don't treat data quality right. Ordinal/Likert data scales cannot be used for algebraic summation. A statistician should prefer other statistics (ordinal regression for example to test differences). T-test and Pearsons correlation isn't appropriate either for these ordinal data.

We did not use the ordinal scale in our study. Our study used indices (such as mean values or sums of scores on Likert-type items) which were treated as quantitative variables. Thus, using such analyses is justified.

Reviewer 2 Report

Impressive work and extensive review of burn-out.

-Why use Democracy index 2107 and not 2020? Is it possible that Poland has declined in democratic values last year. At least the majority of EU-member states think so.

-There is lack of valid refence to the prevalence of insomnia. Figures in Poland authors say is 40-50%. Isn't this too high. A more clinically significant insomnia prevalnece can be found in 10-15% of the population i Europe.

-Authors don't treat data quality right. Ordinal/Likert data scales cannot be used for algebraic summation. A statistician should prefer other statistics (ordinal regression for example to test differences). t-test and Pearsons correlation isn't appropriate either for these ordinal data.

Author Response

Thank you for reviewing our article, Insomnia among prison officers and its relationship with occupational burnout. The role of coping with stress in a Polish and Indonesian sample. Below, we provide information for reviewers about the specific changes we have made to the text, as well as the responses to the reviewers’ comments.

Sincerely,

The Authors

Response to Reviewer 2

  • Presenting results from a cross-sectional study such term as "influence" in the title should be avoided and replaced by "association" or "relationship".

Thank you for this valid remark. Because the study was not experimental in character, using the word “influence” is not justified. In line with your suggestion, the title has been changed to:

Insomnia among prison officers and its relationship with occupational burnout. The role of coping with stress in a Polish and Indonesian sample.

  • The introduction is too lengthy. Please shorten it with the focus on research problems and existing gaps in the literature.

Thank you for this comment. The text has been revised and shortened.

  • Please provide a detailed description of how study participants were recruited. Please explain the reason for recruiting prison officers from different countries. Also, the sample size calculation here is necessary.

A description of the participant recruitment process has been added in the article:

Questionnaires were placed in commonly accessible rooms. The questionnaires were supplemented with a description of the study and its aims, as well as a written request to fill them out and insert them in a specially prepared urn. The description also informed the participants that the questionnaires are entirely anonymous and their results will only be used for the purpose of the current study. The data collection method resulted in a sample which was not random, since the questionnaires were only filled out by interested participants. However, this data collection method was optimal because it allowed for maintaining the participants’ full anonymity and privacy.

Both countries included in the current study were compared in the introduction section. The comparison involves differences in:

-penitentiary systems and executing sentences

-problems related to crime in both countries

-availability of healthcare, including mental healthcare, for prison officers.

It was also indicated that data on insomnia and its relationship with the personal and professional characteristics of prison officers from both countries is missing.

  • A table with Polish and Indonesian samples' characteristics would be very helpful here.

Table 1. Descriptive statistics of surveyed respondents  - added as suggested.

  • References supporting previous validation and structure confirmation of

measures used are not provided. For Burnout Inventory and Coping Orientation subscales, Cronbachs alphas are rather low and some of them are not acceptable. Some data supporting the validity of instruments used for both Polish and Indonesian translations have to be provided.

The following sentences have been added:

-The Polish part of the study used the adaptation of the OLBI by Baka and Basińska (2016). The Cronbach’s α reliability coefficient in the adaptation study was 0.73 for the exhaustion subcale and 0.69 for the disengagement subcale.

-The Indonesian part of the study used the adaptation by Santoso and Hartono (2017). The Cronbach’s α reliability coefficients were 0.798 and 0.753 for the exhaustion and disengagement subscales, respectively.

-The Polish part of the study used the adaptation of the COPE by Juczyński and Ogińska-Bulik (2009). The Cronbach’s α reliability coefficients for the individual subscales ranged from 0.48 to 0.94.

-For the purpose of the Indonesian part of the study, the COPE inventory was translated from English to Bahasa Indonesia, and then back-translated to English. We compared the resulting version of the questionnaire with the English-language original in order to maintain consistency. Thus far in Indonesia, this questionnaire has been used in adolescent and young adult samples.

Reliability testing of this questionnaire revealed that the Cronbach’s α reliability coefficients ranged between 0.5 and 0.9.

The COPE inventory was used in accordance with the information included on the COPE creator’s website (https://local.psy.miami.edu/faculty/ccarver/sclCOPEF.phtml): „You are welcome to use all scales of the COPE, or to choose selected scales for use (see below regarding scoring).  Feel free as well to adapt the language for whatever time scale you are interested in.”

We have added the appropriate references in the text.

Regarding the reliability coefficients, we admit that while they are not high, they are nevertheless acceptable.

  • Please provide a version of the PROCESS macro used and the number of the model(s) applied.

We have added it according to this suggestion.

  • You provide very detailed descriptive statistics for the insomnia scale, but the histogram plot is not necessary and should be removed.

We removed the histogram.

  • Explanations of abbreviations should accompany Table 1. What is D? The tame comment is applied for tables 2-5.

An appropriate legend has been added to the tables, using the following symbols and its meaning:

Sc.                  Skewness

D                    Kolmogorov-Smirnov test statistics

r                      Pearson’s correlation coefficient

Z                     Fisher’s Z test statistics

r̅                      average rank

IQR                interquartile range

P                     significance level

β                     standardized coefficient b

  • Please provide a reference for insomnia cut-offs.

The article contains a reference to the literature in the following sentence: The original criteria consider scores above six as reliable indicators of early symptoms of insomnia [104,105].

  • In Figure 8 some path coefficients are very low (e.g., for age and sex). Please indicate statistical significance in the figure. Also, I have some doubts about categorical variables (like gender) analysis with the AMOS, I think there is a problem here. You should use other software if you want to include categorical predictors like MPlus or lavaan in R (free). Next, some variables like type of prison and multishift-work system appear in the final path model. These variables are not presented in the methods section. The same comment applies if these two variables are categorical.

The information available at  https://www.ibm.com/support/pages/binary-variables-amos contains an explanation regarding the possibility of using binary variables in AMOS. Using additional software is not necessary.

The study involved the following variables: type of prison and multi-shift work.

  • Overall, the paper is too lengthy. The authors should concentrate the information in the introduction, results description and discussion and reduce paper length.

This is a valid remark. We have revised the text according to your suggestion.

Round 2

Reviewer 1 Report

Dear Authors,

Thank you for the improved version of your paper. Corrections made are adequate except for Fig. 8. I recommend consulting a professional.  Path analysis and equation modelling should be applied in case of testing direct and mediated effects between study variables. I would suggest changing this model by simple linear regression as all the variables were entered in one step and no mediation analysis was conducted. This really detracts from all the nice work done.
In the statistical analysis section, please provide the number(s) of the models applied (models as a parameter in the PROCESS function) as in version PROCESS macro 3.5 there are 92 models available.

Author Response

Reply to the Review Report

In line with the Reviewer's suggestion, we have consulted the correctness of our analyses and their relation to the assumptions of our research. We have obtained information which confirms the validity of our prior statistical analyses:

Structural equation modeling (SEM) is used not only for mediation.
The analysis indicated by the Reviewer included both moderation and so-called "multigroup analysis" which allows for comparing path analyses between the countries. A simple regression analysis does not permit such comparisons because it assumes "one dependent variable-one independent variable" relationships -> regarding the suggested PROCESS macro.
The model in our study was more complex. Thus, we used a more complex statistical method based on http://web.pdx.edu/~newsomj/semclass/ho_moderation.pdf
Regarding the PROCESS macro, naturally it includes many possible models for testing. However, it nevertheless contains models of the "one variable X, one variable Y, maximum two moderators" type.
The aim of our analysis was to show general differences rather than individual paths in separation, without any links. In short, using the PROCESS macro would not allow for simultaneously including the three burnout dimensions and the control variables.

Authors